# Eco-Friendly Castor Oil-Based Composite with High Clam Shell Powder Content

**DOI:** 10.3390/polym16233232

**Published:** 2024-11-21

**Authors:** Fangqing Weng, Kui Jian, Yazhou Yi, Peirui Zhang, Ernest Koranteng, Qing Huang, Jiahui Liu, Guoping Zeng

**Affiliations:** 1Hubei Key Laboratory of Purification and Application of Plant Anti-Cancer Active Ingredients, College of Chemistry and Life Sciences, Hubei University of Education, Wuhan 430205, China; wengfq@hue.edu.cn (F.W.); yellowgreenyellow@hotmail.com (Q.H.); maijv129813@yeah.net (J.L.); 2College of Chemistry, Central China Normal University, Wuhan 430079, China; kjian@mails.ccnu.edu.cn (K.J.); yiyazhou@mails.ccnu.edu.cn (Y.Y.); peiruizhang@mails.ccnu.edu.cn (P.Z.); 3Department of Chemistry Education, University of Education, Winneba P.O. Box 25, Ghana; ekoranteng@uew.edu.gh

**Keywords:** high biomass content, castor oil, clam shell, composites

## Abstract

Eco-friendly castor oil-based composites with a high content of clam shell powder were prepared in this study. Biomass composites were prepared by blending castor-oil-based polyurethane prepolymer (COPU) with a filler consisting of high-content clam shell powder (CSP), named CSP-COPU. The structure, microstructure, mechanical properties, and thermal stability of the composites were investigated. The results showed that even at a loading as high as 75 wt.% of the CSP filler, the composite still exhibited good tensile strength and elongation at break. Furthermore, compared with the CSP-COPU composites, TCOS-50 synthesized through blending OH-terminated castor oil-based polyurethane prepolymer (TCOPU) and CSP filler proved that the chemical bond between COPU containing terminal -NCO groups and CSP containing active -OH groups was the key reason to obtaining the composite material with desirable properties. These findings provide prospects for applying biomass-loaded CSP-COPU composites in the packaging industry while contributing to carbon peak achievement and carbon neutrality.

## 1. Introduction

Current global concerns, including the depletion of oil resources, the effects of global warming, and environmental sustainability [1,2,3], require us to study and utilize renewable energy sources in particular. Consequently, vegetable oils have become one of the ideal raw materials for synthesizing polymers because they are non-toxic, non-hazardous, and renewable [4,5]. Among the many vegetable oils, castor oil (CO) has been employed directly in preparing polyurethane coatings, elastomers, adhesives, foams, and interpenetrating polymer networks [6,7]. CO-based polyurethane networks exhibit attractive flexibility and water resistance properties [8]. For many years, castor oil-based polyurethane and castor-oil-based polyurethane prepolymer (COPU) with good water resistance and flexibility have been prepared by using castor oil and isocyanate [9,10,11,12]. These COPUs have been utilized as compatibilizers [8], flexibility modifiers [11], and adhesives [12] for the preparation of composite materials. For instance, COPU can function as a modifier to improve a material’s flexibility mainly because of its flexible long chains [8,9,10,11,12]. In addition, by adjusting the ratio of reactants (castor oil and isocyanate), CO-based polyurethane prepolymers with reactive groups (-NCO) can be prepared to be used as compatibilizers, which can be further manipulated to form chemical bonds with other polymers to give valuable materials [13].

To this end, CO-based polyurethane materials have been utilized to modify various polymeric materials, such as ethoxylated prepolymer that has been synthesized using isophorone diisocyanate (IPDI) and polyethylene glycol (PEG) [14]. The synthesized prepolymer was further reacted with castor oil to prepare castor oil-based polyurethane as a ligand thickener. What’s more, the phenolic foam was modified with castor oil-based polyurethane prepolymer to improve the material’s flexural strength [15]. In addition, the castor oil-based polyurethane prepolymer was also used as a compatibilizer to obtain a toughened and reinforced thermoplastic starch-based material [8]. Despite these remarkable achievements, castor oil-based polyurethane materials are still unsuitable for structural and high-performance composite applications due to their inherently poor mechanical properties [8,9]. Therefore, this work focuses on using COPU to prepare high-strength composite materials while using biomass raw materials entirely.

Many studies have shown that adding fillers to polymer materials is one of the simplest and most effective methods to obtain reinforced polymer materials. Some of the commonly used fillers include calcium carbonate (CaCO_3_), barium sulfate (BaSO_4_), silica, etc. [16,17]. For instance, CaCO_3_ was reported to effectively improve the storage modulus and thermal decomposition temperature of maleated castor oil-based composites [18]. The challenge in utilizing these fillers is that due to their weak interaction with the matrix, interface incompatibility often occurs in the resulting composite materials [16,17]. Furthermore, the advent of natural fillers and their substitution for conventional mineral or petroleum-based fillers has emerged as an appealing approach, driven by the scarcity of petroleum resources and the context of sustainable development [19,20,21]. Therefore, this work aims to solve the problem of interfacial incompatibility in composites prepared using common inorganic fillers through simple and readily available natural fillers. The interaction between the natural filler and the matrix via chemical bonding is expected to significantly enhance the composite’s interfacial compatibility to afford high-performance composite material.

The presence of seashells provides an eye-catching avenue for this work. As common biomass, shells are eco-friendly and renewable resources that are inexpensive and readily available but were once considered as waste with a substantial environmental threat [22]. Interestingly, shells encompass a minute quantity of polysaccharide protein, which features an organic structure that includes several active groups, such as hydroxyl, amino, etc. [23,24,25]. Furthermore, shells exhibit a layered, brick-and-mortar stacking structure, which endows them with exceptional mechanical properties [26,27]. Additionally, seashells are made up of about 95% CaCO_3_, which makes an outstanding contribution to the hardness exhibited by the material [22]. Therefore, seashells are ideal natural biofillers to enhance the performance of COPU.

Based on the above background, in this work, we further take full advantage of readily available renewable resources, such as clam shells and castor oil, to investigate the structure and performance of castor oil-based thermosetting composites with high biomass content. We envisaged preparing castor oil-based thermosetting composites with high biomass content by adjusting the content of shell powder fillers and the COPU matrix. We also prove the importance of the active functional groups of COPU in preparing thermoset composites. It is desirable to produce composites with high biomass content and good tensile properties for application in the packaging industry and help to achieve carbon peak as well as carbon neutrality.

## 2. Experimental

### 2.1. Materials

Castor oil (CO, *M_w_* = 929 g mol^−1^) was provided by Sinopharm Chemical Reagent Co., Ltd. (Shanghai, China). 4, 4′-Diphenylmethane diisocyanate (MDI) was produced by Sigma Corporation (Shanghai, China). Clam shell powder (CSP) was self-made in the laboratory, referring to previous work [27].

### 2.2. Synthesis of Castor Oil-Based Polyurethane Prepolymers

According to our previous work [26,27], NCO-terminated castor oil-based polyurethane prepolymer (COPU) was synthesized by MDI and castor oil with a -NCO to -OH molar ratio of 2:1. Then the obtained COPU was taken in a frame plate with a poly tetra fluoroethylene film at 160 °C under the condition of 10 MPa for 4 h to obtain the COPU sheet [11,12].

OH-terminated castor oil-based polyurethane prepolymer (TCOPU) was synthesized with a molar ratio of -NCO to -OH of 1:2, and the synthesis procedure was the same as that of COPU. The IR image of TCOPU is available in Appendix A.

### 2.3. Preparation of Castor Oil-Enhanced Polyurethane Prepolymer Materials with Different Contents of Clam Shell Powder

Different contents of CSP were directly added to the newly synthesized COPU at 90 °C and stirred for 20 min at 90 °C. It is worth noting that the viscosity of COPU increased significantly after temperature reduction, which is not conducive to the uniform blending of CSP and COPU. The synthesized material was then immediately poured into a hot-pressed frame plate and hot-pressed at 160 °C and 10 MPa for 4 h to obtain thermoset composites. During this process, the mass percentages of CSP added were 25 wt.%, 50 wt.%, and 75 wt.%, and the resulting composites were named COS-25, COS-50, and COS-75, respectively, shown in Figure 1a.

Using the same experimental method, the composite prepared with 50 wt.% TCOPU and 50 wt.% CSP was named TCOS-50 (shown in Figure 1b). The specific formula is shown in Table 1.

### 2.4. Characterization

The Fourier Transform Infrared (IR) spectrum of the composite samples at the range of 4000~400 cm^−1^ was recorded using an IR spectrometer (Nicolet6700, Thermo Fisher Scientific, Waltham, MA, USA).

The fractured surface of the composite at a voltage of 20 kV was observed by a scanning electron microscope (SEM, JSM-5610LV, JEOL Ltd., Tokyo, Japan).

X-ray diffractograms (XRD) of the samples were recorded at 2θ = 10~80° with a scanning rate of 6°/min and cathode at 40 kV and 40 mA through an X-Ray Diffractometer (D8 Advance, Bruker Instruments LTD., Beijing, China).

The fracture strength (σ_b_, MPa), elongation at break (ε_b_, %), and elastic modulus (E, MPa) of the composites were determined with a GP-TS2000S universal testing device (Shenzhen high-quality testing equipment Co., LTD, Shenzhen, China) according to the GB/T1040.3-2006 standard [28]. 

The thermogravimetric analysis (TGA) was conducted using a TGA STA449C thermogravimetric analyzer (Neichi Scientific Instrument Trading (Shanghai) Co., LTD. (China Headquarters)), heating from room temperature to 900 °C within a nitrogen atmosphere.

The hydrophilicity of the film surface of the composite was assessed by measuring the water contact angle (CA) with an OCA20 goniometer (DataPhysics, Filderstadt, Germany), employing the static sessile drop technique.

## 3. Results and Analysis

### 3.1. Structural Analysis of COS Composites

To further reveal the differences in chemical composition between CSP and CaCO_3_, their infrared spectra are compared. For the CS sample in Figure 2a, the strong absorption peak at approximately 3400 cm^−1^ overlaps with the -NH stretching vibration of polysaccharides or proteins (the -NH absorption band at 3260 cm^−1^), while the peak around 3000 cm^−1^ is attributed to the stretching vibration of hydroxyl groups (-OH). Additionally, the peak at 1740 cm^−1^ indicates the stretching vibration of carbonyl groups (C=O). We also observed N-H stretching vibrations at 1240 and 1510 cm^−1^. In contrast, no absorption peak for -NH was found in the spectrum of pure CaCO_3_. Figure 2b,c show the IR spectra of COPU and COS-25, COS-50, and COS-75 at 24 h and on the 16th day. From Figure 2b, the characteristic peak induced by the N-H bond stretching vibration in carbamate for COPU, COS-25, and COS-50 samples appeared at 3310 cm^−1^, while the C=O stretching vibration peak in the ester group occurred around 1720 cm^−1^ all within 24 h. As mentioned earlier, the results are because the terminal -NCO groups in COPU can interact with -OH and -NH_2_ groups to form carbamate bonds, etc. Interestingly, the spectra for COPU, COS-25, and COS-50 in Figure 2b show a prominent -NCO peak at about 2267 cm^−1^ [29,30,31]. However, the -NCO peak decreases gradually as the shell powder content increases. These phenomena indicate that shell powder tends to consume the terminal -NCO group in COPU because the organic components of polysaccharide-protein in shell powder contain active groups (such as -OH, -NH-, etc.), which can react with the -NCO group [31]. This observation is consistent with previous research conclusions [26].

After 16 days, the spectra shown in Figure 2c indicated that the -NCO characteristic peak for all the composite weakened even though there are still relatively prominent peaks for COPU and COS-25. The characteristic absorption peak of the -NCO group for COS-50 could hardly be seen again because the active groups in shell powder could gradually consume the -NCO functional groups in COPU [26]. Meanwhile, COPU could only rely on moisture in the air to slowly consume the terminal -NCO groups. It is important to note that because the shell powder content in COS-25 is relatively small, the consumption rate of -NCO is slower than that of COS-50. As a result, the absorption peak of COS-50 at 2267 cm^−1^ was significantly weaker than that of COS-25. The infrared spectrogram results show that the shell powder reacts with COPU to consume the -NCO and generate a urethane bond [26,31].

Figure 3 shows the SEM images of the composites at different magnifications (×500, ×3000). It can be seen from the figures that the shell powder mixed well with COPU and was uniformly dispersed in the COPU matrix. At a higher shell powder loading (about 75 wt.%), a small amount of the shell powder agglomerated with a small amount of the COPU continuous phase. By comparing Figure 3(A1,B1,C1) it is evident that with increasing shell powder content, the surface became rough, giving a rough, folded surface (for example the red circle in Figure 3(C1)). This observation was mainly because shell formation is a complex biomineralization process and usually makes the shell powder show a brick-cement structure at the microscopic level [32].

In addition, it can be seen from Figure 3(A2,B2,C2) that there are almost no cracks or gullies between the two phases on the fractured surface of composite materials, indicating that the compatibility between shell powder and COPU is relatively desirable. As already mentioned, the improved compatibility could be attributed to the reaction between the active groups (such as -NH_2_ or -OH, etc.) in the organic matter of shell powder and the -NCO group of COPU to produce a urethane bond [26], as confirmed by the infrared image. However, the increase in shell powder content makes the fractured surface of the material more uneven.

Figure 4 shows the XRD pattern of the CSP and COPU-based composites. It can be seen from the figure that the shell powder is still in the state of aragonite (PDF#41-1475) [33,34,35] after hot-pressing and curing. The crystallization peak of the composite is highly similar to that of the original shell [26]. As the content of shell powder increases, the peak strength of the material is slightly increased, the trend of XRD of different samples can be observed with the guidance of the arrow, but the crystallization property does not change while the crystallization property of the shell powder is enhanced.

### 3.2. Analysis of Basic Properties of COS Composites

Table 2 shows the data for the tensile property of COPU, COS-25, COS-50, and COS-75, including tensile strength, elastic modulus, and elongation at break. The pure COPU displayed a tensile strength of 22.9 MPa, elastic modulus of 2036.6MPa, and elongation of 40.3%, indicating that COPU is a ductile material [36]. After adding 25 wt.% of shell powder filler, the tensile strength of COS-25 increased significantly to 32.2 MPa, which is an increase of 40.6%. However, the elastic modulus of COS-25 increased rapidly from 2036.6MPa of pure COPU to 3520.9 MPa, representing an increase of nearly 1.5 GPa. The elastic modulus of COS-25 is about 1.7 times the elastic modulus of COPU, although the elongation reduced significantly to 4.5%. In addition, after adding 50 wt.% of shell powder filler, the tensile strength of COS-50 increased to 35.3 ± 1.2MPa, representing an increase of 54% compared with pure COPU. The elastic modulus of COS-50 increased sharply to 5859.0MPa, representing an increase of nearly 4 GPa and about 2.9 times the elastic modulus of COPU. The results show that COPU transformed from a flexible soft polymer material into a hard high-strength material. However, after adding 75 wt.% of shell powder filler, the tensile strength reduced to 8.1 MPa, while the elastic modulus and the elongation at break reduced to 4160.3 MPa and 11.1%, respectively. The decrease in strength and modulus may be caused by the decomposition of polyglycoproteins in shell powder and the deactivation of the COPU terminal groups during hot-pressing time. All these phenomena indicated that CSP and COPU did not have only physical interactions but chemical interactions to form chemical bonds [26,31,37]. As a result, the COS-50 sample exhibited higher tensile strength and elastic modulus.

In addition, from Figure 1, the blending of TCOPU and CSP to obtain TCOS-50 could not occur under the same blending conditions for the COS-50 composites. The reason for such an observation is that there is no -NCO group in TCOPU to interact chemically with the active groups in CSP, making it difficult to form a thermoset/thermoplastic in the mold let alone be utilized for further applications. The observation further proved that the chemical bond between COPU (-NCO) and CSP (-OH) is the key to obtaining a composite material with desirable properties.

Figure 5 shows the thermal stability of COPU composites measured in a nitrogen atmosphere. As illustrated in Figure 5, the decomposition process of composite materials can be categorized into two distinct stages. The temperature range in the first stage is about 250 °C~550 °C due to the thermal decomposition of the soft segment structures in COPU, which is consistent with the temperature range reported by Wu et al [8]. The weight loss of the composite materials, which represents the second stage of decomposition, has a temperature range of 650~800 °C corresponding to the decomposition of CaCO_3_ into lime (CaO) [26,31]. The rate of weight loss for COS-25, COS-50, and COS-75 is 11.54%, 21.30%, and 31.50%, respectively.

In terms of the residue for the composites, COPU, COS-25, COS-50, and COS-75 had 7.00%, 15.30%, 28.29%, and 42.40% as the final residues at 900 °C, respectively. The residual amount of COPU was carbon, and the residual amounts of COS composites were CaO and carbon (the specific TG data are shown in Table 3) [8,26,31]. The thermal stability of the composite was not negatively affected by the increase of shell powder content.

In general, the more hydrophilic the material is, the lower the value of its contact angle [38]. Figure 6 shows the water contact angle of COPU, COS-25, COS-50, and COS-75 composites. The contact angle of pure COPU was 102.1°, but after adding 25 wt.% and 50% of shell powder filler, the contact angle of the obtained composites of COS-25 and COS-50 increased, reaching 106.0° and 108.9°, respectively. This means that the hydrophobic property of the samples was further improved, indicating that the organic groups on the surface of the shell powder were effectively consumed. These results confirmed the infrared and mechanical properties and SEM tests. However, when the shell powder content was increased to 75 wt.%, the contact angle of COS-75 dropped to 96.5° because the content of the hydrophilic component far exceeded that of the hydrophobic component. Nevertheless, the composite was still a hydrophobic material. It is worth noting that the consumption of the active groups of the organic matter in shell powder via a chemical reaction with COPU plays a vital role in improving the hydrophobicity of the composite material. Therefore, control of the various proportions of the components in the composite is critical. For instance, better and stronger interface interaction is achieved to improve the hydrophobicity of the materials when the mass fraction ratio of CSP and COPU is 1:1.

### 3.3. Principle Analysis of COS Composite Materials

Figure 7 shows the schematic diagram of COS-50 composite material. The active surface of shell powder can interact with the castor-oil-based polyurethane prepolymer polymer chain to form a cross-linked structure, further enhancing the composite material. Due to the strong interfacial interaction between the filler particles and the polymer matrix in COS-50, the material can withstand more significant stress and avoid damage. At the same time, the flexible polymer chain combined with shell powder can absorb energy by changing the molecular conformation so that the composite material’s tensile strength and elastic modulus can be improved [39].

## 4. Conclusions

An eco-friendly renewable shell powder was proposed as a filler material, and the shell powder was also serves as reinforcing agent to endow castor oil-based polyurethane with better tensile properties. Subsequently, a cross-linked castor oil-based polyurethane composite material was successfully prepared by the hot-pressing curing method. To this end, the effects of shell powder content on the interfacial structure and properties of COPU composites were investigated. The overall results showed that as the shell powder content reached 50 wt.%, the tensile strength of COS-50 increased by 1.5 times that of pure COPU, while elastic modulus increased by 2.9 times, close to 6GPa. Thus, COS-50 exhibited excellent interface compatibility, the best tensile properties, and water resistance. In addition, the raw materials of the composite, namely castor oil and shell powder, are renewable biological materials and fall in line with the demand for low carbon material for a sustainable environment. The comprehensive results indicated that the new castor oil-based polyurethane composites with high biomass content have a potential industrial application in structural (support) materials.

## Figures and Tables

**Figure 1 polymers-16-03232-f001:**
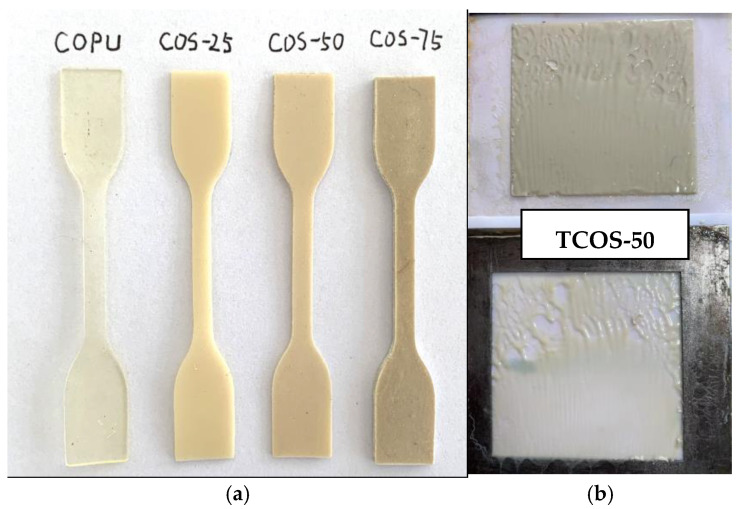
(**a**) Composite samples of COPU, COS-25,.COS-50, COS-75; (**b**) Composite samples of TCOS-50.

**Figure 2 polymers-16-03232-f002:**
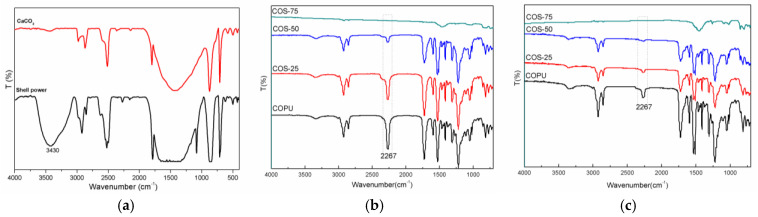
(**a**) IR image of clam shell powder; IR image of COS composite material (**b**) placed for 24 h and (**c**) placed for 16 days.

**Figure 3 polymers-16-03232-f003:**
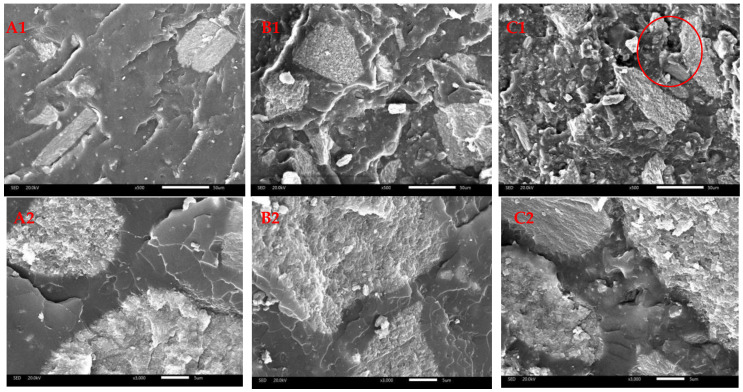
SEM images of (**A1**) COS-25 (×500), (**A2**) COS-25 (×3000), (**B1**) COS-50 (×500), (**B2**) (×3000), and (**C1**) COS-75 (×500), (**C2**) COS-75 (×3000).

**Figure 4 polymers-16-03232-f004:**
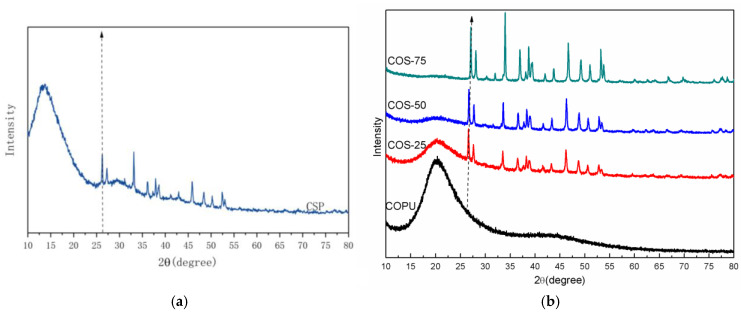
XRD pattern of (**a**) CSP and (**b**) COPU and composites.

**Figure 5 polymers-16-03232-f005:**
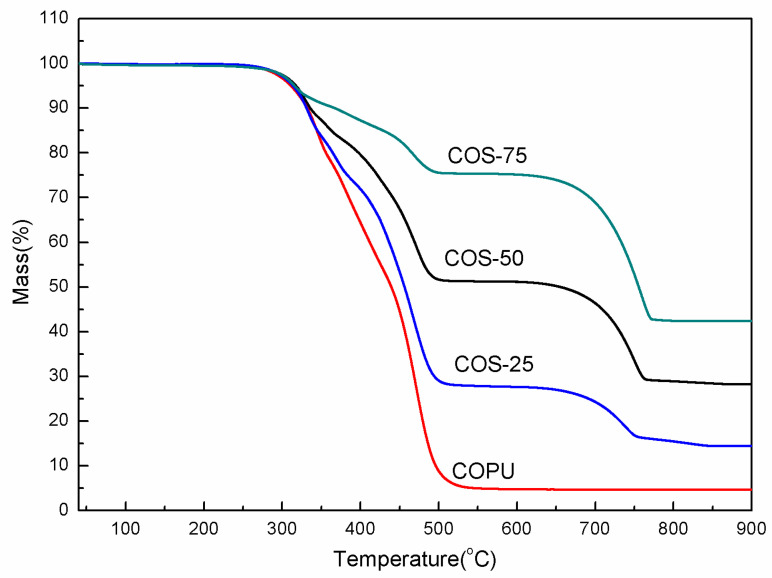
TG diagram of COPU and COS composites.

**Figure 6 polymers-16-03232-f006:**
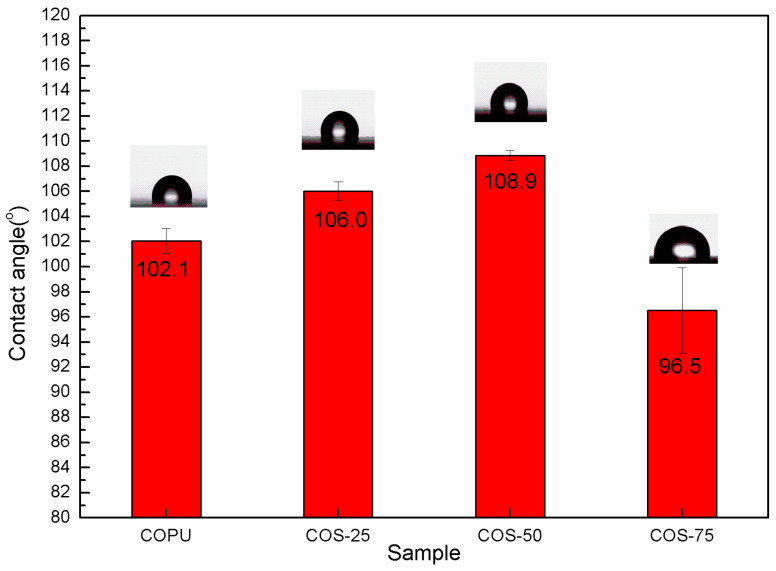
Contact angle diagram of COS composites.

**Figure 7 polymers-16-03232-f007:**
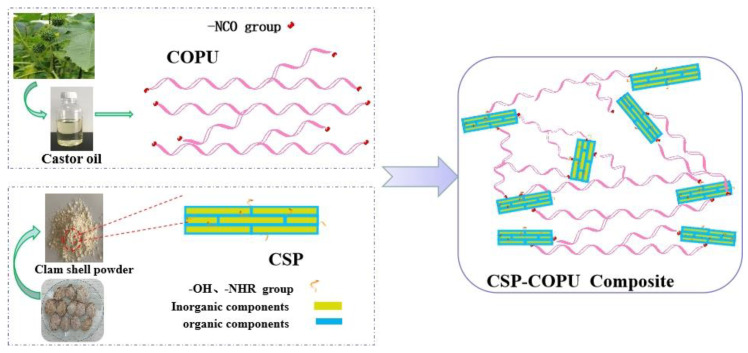
Interface structure of COS-50 composite material.

**Table 1 polymers-16-03232-t001:** The formulas and the sample names of the composites.

Sample Names	COPU (wt.%)	CSP (wt.%)	TCOPU (wt.%)
COPU	100	0	-
COS-25	75	25	-
COS-50	50	50	-
COS-75	25	75	-
TCOS-50	-	50	50

**Table 2 polymers-16-03232-t002:** Tensile properties of the composites.

Sample	Tensile Strength δ_b_ (MPa)	Elastic Modulus *E* (MPa)	Elongation at Break ε_b_ (%)
COPU	22.9 ± 1.4	2036.6 ± 196.9	40.3 ± 7.0
COS-25	32.2 ± 0.7	3520.9 ± 146.2	4.5 ± 0.9
COS-50	35.3 ± 1.2	5880.7 ± 147.8	4.0 ± 0.6
COS-75	8.1 ± 0.1	4160.3 ± 31.3	11.1 ± 0.9

**Table 3 polymers-16-03232-t003:** Weightlessness and ash ratio at 900 °C for the two main stages of different composites.

Sample	Weight Loss During 250~550 °C (%)	Weight Loss During 650~800 °C (%)	Ashes at 900 °C (%)
COPU	9.54	0.033	7.00
COS-25	71.79	11.54	15.30
COS-50	48.06	21.30	28.29
COS-75	23.87	31.50	42.40

## Data Availability

The data presented in this study are available on request from the corresponding author because our research group is still conducting further explorations related to this study.

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
