# Peer review of "Eco-Friendly Castor Oil-Based Composite with High Clam Shell Powder Content"

_polymers, 2024, doi:10.3390/polym16233232_

Round 1
Reviewer 1 Report
Comments and Suggestions for Authors
Composites were prepared using a castor-oil based polyurethane and clam shell powder. The ratio of shell was varied to determine its effect on properties. This study is interesting, however it lacks much quantification and understanding of the properties in its current state. A series of revisions are recommended to strengthen the overall work and make it appropriate for publication.
1. Please check your English grammar throughout the manuscript.
2. Make sure you are labeling all abbreviations after their first use. For example, TCOS-50 is used in the abstract without any definition.
3. Please expand on the use of seashells as fillers in composite materials. There should be literature which explains how the presence of surface functional groups can specifically strengthen a composite.
4. In the experimental section, “The clamshell powder (CSP) was self-made in the laboratory, referring to previous work.” Please add the references for which you are referring to.
5. On page 3, “It is worth noting that the viscosity of COPU decreased significantly after temperature reduction, which is not conducive to the uniform blending of CSP and COPU.” Please clarify. Viscosity should increase with lower temperature.
6. Please fix the labeling in Figure 1.
7. Please explain why such a low ratio of isocyanate to hydroxyl group was chosen for these composites. Additionally, please revise the discussion on FTIR of the composites. Phrasing throughout this section is confusing and is presented wrong in a few cases.
8. Please fix Figure 3. The labels and scale bars are not easily visible.
9. Figure 3, C1 and C2 are listed as having a “rough and folded surface”. This is not seen in the imaging, please revise or choose better imaging.
10. Was XRD collected on only the shell itself? This could be useful to add.
11. The elastic modulus for COS-75 was listed as both 416.3 MPa and 4160.3 MPa, please choose the correct value and fix it throughout the work. Additionally, the explanation provided for the decrease in performance of COS-75 does not make sense. The presence of additional chemical bonds would strengthen tensile performance, not lessen it.
12. The discussion on TCOS-50 on page 7 seems out of place. There is no characterization connected to this section, please add any data or relocate this discussion.
13. Please add to the TGA discussion that the initial decomposition is consistent with the COPU content.
14. The drop in water contact angle for COS-75 is not property explained. According to FTIR, the surface chemical groups were consumed…so that cannot be the explanation for a decrease in water contact angle. Please rethink your explanation for this data.
15. Section 3.3 seems unnecessary and could simply be added to the conclusion.
16. The surface chemistry on the CSP is obviously very important to the performance of the composites, but no attempt was made to quantify the actual chemistry. This is important and should be fixed. Even just a simple FTIR would provide ample information to support your claims.
Comments on the Quality of English LanguageThe english needs to be improved, especially in the introduction, to allow for a full understanding of the work
Author Response
Reviewer1
- Please check your English grammar throughout the manuscript.
Answer : We have proofread the full English grammar, thank you for your suggestion.
- Make sure you are labeling all abbreviations after their first use. For example, TCOS-50 is used in the abstract without any definition.
Answer : We indeed recognize this as an oversight on our part, which has since been thoroughly reviewed and enhanced.
TCOS-50 refers to the comoposite that synthesized through blending OH-terminated castor oil-based polyurethane prepolymer (TCOPU) and CSP filler.
- Please expand on the use of seashells as fillers in composite materials. There should be literature which explains how the presence of surface functional groups can specifically strengthen a composite.
Answer : As a common and abundant natural resource, shells have long been treated as solid waste in coastal areas, which does not only constitute a waste of natural resources but also as a pollutant to the environment. However, this ubiquitous agricultural waste is cheap, easy to obtain, environmentally friendly, and has been found to be a natural polymer complex structure containing proteoglycan in addition to calcium carbonate (CaCO3) Interestingly, seashells contain a small amount of polysaccharide-protein with an organic structure containing some active groups, such as hydroxyl, amino, etc. Additionally, seashells are made up of about 95% CaCO3, which makes an outstanding contribution to the hardness exhibited by the material. Moreover, shells have layered stacking structure in the form of “Brick-mortar”, thus possessing outstanding mechanical properties. The combination of COPU with the shell powder will not only reduce the cost but also improve the mechanical strength of the material.
We also appropriately supplemented the introduction of shell powder in the article.
- In the experimental section, “The clamshell powder (CSP) was self-made in the laboratory, referring to previous work.” Please add the references for which you are referring to.
Answer : Yes, thank you for the reminder, we have cited the reference now, which is reference number 27.
- On page 3, “It is worth noting that the viscosity of COPU decreased significantly after temperature reduction, which is not conducive to the uniform blending of CSP and COPU.” Please clarify. Viscosity should increase with lower temperature.
Answer : I'm truly sorry, this was not our intention and was an error that should not have occurred. The error has been promptly corrected. Namely, “It is worth noting that the viscosity of COPU increased significantly after temperature reduction...”
- Please fix the labeling in Figure 1.
Answer : Thanks for your reminder, we have changed it to“Figure 1.Composite samples with different amount of CSP.”Furthermore, due to the inability of TCOS-50 to achieve solidification and molding, it was not possible to produce dumbbell bar samples, and therefore it was displayed in its actual form.
- Please explain why such a low ratio of isocyanate to hydroxyl group was chosen for these composites. Additionally, please revise the discussion on FTIR of the composites. Phrasing throughout this section is confusing and is presented wrong in a few cases.
Answer : Thank you for your detailed review. The choice of a ratio of isocyanate to hydroxyl groups of 2:1 is made to prepare COPU with -NCO groups at the ends. References 8, 26, 27, and 30 all use this ratio for the synthesis of polyurethane prepolymers. According to the reports from these references, the resulting polyurethane prepolymers have good reactivity. Therefore, this study also adopts the same ratio. As for the analysis of infrared, we have also made corrections, and supplemented with IR image of clam shell powder and CaCO3.
- Please fix Figure 3. The labels and scale bars are not easily visible.
Answer : Dear experts, in fact, each SEM image is equipped with a scale bar at the bottom right corner, where the scale bars for images A1, B1, and C1 is 50μm, and for images A2, B2, and C2, it is 5μm (as shown in the following figure).You might be able to see them more clearly if you zoom in on the page a bit. We apologize for any inconvenience this may have caused.
- Figure 3, C1 and C2 are listed as having a “rough and folded surface”. This is not seen in the imaging, please revise or choose better imaging.
Answer : Thank you for your careful observation, in the cross-sections of C1 and C2 (as shown in the red boxes in the figure), the rough and folded surface is actually relative to A1, A2, and B1, B2. We have indicated this with red boxes in the following figure and have also made a new selection of figureC while respecting the facts.
- Was XRD collected on only the shell itself? This could be useful to add.
Answer : Thank you for your suggestion, we have added the XRD data of the shell powder and conducted an analysis.
- The elastic modulus for COS-75 was listed as both 416.3 MPa and 4160.3 MPa, please choose the correct value and fix it throughout the work. Additionally, the explanation provided for the decrease in performance of COS-75 does not make sense. The presence of additional chemical bonds would strengthen tensile performance, not lessen it.
Answer : Thank you for your correction, the correct value should be 4160.3 MPa, and it has been corrected.
As for the tensile strength of COS-75, in principle, due to the inherently high strength of CSP, theoretically increasing the content of CSP should enhance the tensile strength of the resulting material. However, more crucially, when the content of CSP is excessively high, while the amount of COPU is also limited (that is, the reactive groups capable of reacting with CSP), the excess of CSP cannot form effective interactive forces with the matrix. Thus the interfacial compatibility issue between components has not been properly addressed, this will actually lead to a reduction in tensile strength.
- The discussion on TCOS-50 on page 7 seems out of place. There is no characterization connected to this section, please add any data or relocate this discussion.
Answer : Indeed, Figure 1 presents the current state of TCOS-50. Due to the lack of functional groups(-NCO) in the TCOPU that can react with CSP, even under the same experimental conditions, the two only exhibit physical mixing without forming a composite material with a stable morphology. Therefore, we are unable to measure its mechanical properties. To further support this, we also provide the infrared spectrum of TCOPU, in which there is no characteristic peak of the -NCO group (around 2270cm-1). For details, please refer to the figure below:
- Please add to the TGA discussion that the initial decomposition is consistent with the COPU content.
Answer : In the initial stage, the decomposition trend of the composite material is similar to that of COPU, which is mainly attributed to the decomposition of the soft segment structures in both the composite material and COPU.And although the shell powder forms urethane bonds with COPU, these chemical bonds are of the same type as some of those existing within COPU itself. However, it is noteworthy that there is a significant difference in the degree of their decomposition (residual amount). Moreover, after 500 degrees, the decomposition trend of COPU is markedly different from that of the composite material.
- The drop in water contact angle for COS-75 is not property explained. According to FTIR, the surface chemical groups were consumed…so that cannot be the explanation for a decrease in water contact angle. Please rethink your explanation for this data.
Answer : Yes, through infrared spectroscopy, we can indeed find that the -NCO is ultimately consumed by the -OH. However, when the content of shell powder reaches as high as 75%, the tensile property data indicates that the content of COPU in COPU-75 is not sufficient to consume all the hydrophilic groups in the shell powder. From the perspective of stoichiometry, it is clear that CSP is already in excess at this point. Therefore, we found that the contact angle of COS-75 has decreased through contact angle tests, which means the hydrophilicity has increased. This is precisely corroborated by the previous IR and tensile property data.
- Section 3.3 seems unnecessary and could simply be added to the conclusion.
Answer : Thank you for your suggestion; if the journal format permits, we would also be willing to adjust this part to the conclusion section.
- The surface chemistry on the CSP is obviously very important to the performance of the composites, but no attempt was made to quantify the actual chemistry. This is important and should be fixed. Even just a simple FTIR would provide ample information to support your claims.
Answer : Thank you, your suggestion makes a lot of sense to us. We have added the infrared spectra of shell powder and calcium carbonate for better analysis and support.

Reviewer 2 Report
Comments and Suggestions for Authors
The main objective of the authors in this manuscript is to obtain environmentally friendly composite materials basis on a polyurethane polymer from castor oil and clam shell powder as filler. The manuscript is addressed to readers interested in obtaining polymer materials from natural renewable resources and their characterization. The paper is written in four main parts including: introduction, experimental part, obtained results and final conclusions. At the beginning of the manuscript there is a short summary in which the authors make a brief presentation of the main aspects of the research initiated by the authors.
The summary is clearly written and provides a picture of the essence of the paper. However, I would recommend the authors to stop using abbreviations in this section because they make it difficult to follow the aspects presented in the work. Abbreviations can be used as needed in the other sections of the manuscript with the meaning defined immediately after use.
In the introduction section, the authors critically analyze the literature of recent years dedicated to the use of castor oil together with different diisocyanates and diols to obtain polyurethanes with improved flexibility. At the same time, the negative effects of the presence of castor oil on the other mechanical properties are highlighted, which limits the use of these polymers in structural applications; A simple solution taken from specialized literature to improve the mechanical characteristics of polymers is the creation of composite materials reinforced with inorganic particles such as calcium carbonate, barium carbonate, silica, etc. Carefully following the published data, the authors found that the presence of CaCO3 particles improves the mechano-dynamic and thermal properties of polymer composites. Starting from the presented aspects, the authors proposed in this paper to obtain composite materials from thermocrosslinkable polyurethane matrix based on castor oil and calcium carbonate particles of organic origin, such as the powder of clam shells. Although the idea of ​​using the powder from clam shells to obtain composite materials together with synthetic polymers is not really new, this approach is still interesting.
The second section of the manuscript is dedicated to experimental aspects. Both the materials used and the laboratory and the techniques for the preparation of the polyurethane prepolymer with castor oil content, the shell powder, as well as the preparation several composite made with the two types of materials are clearly described. The equipment used for characterization is quite well presented.
Data regarding the structural characterization of the obtained composites as well as the properties such as crystallinity, the behavior of the composites under mechanical stress, the morphological aspects are well described in the "Results" section. The thermograms in Figure 5 should also be completed with the derived curves. The stages of thermal degradation will be better highlighted. Other data such as the starting temperature of the thermal degradation (T5%), the temperature at which the thermal degradation occurs at the maximum speed (Tm) and the temperature that characterizes the end of the thermal degradation (Tf) also be filled in the manuscript and commented according with the filler content. The conclusions are well supported by experimental data. The selected bibliography supports the authors' approach regarding the chosen research direction. The publications are up-to-date and an exaggerated number of self-citations were not found.
Author Response
Reviewer 2
The main objective of the authors in this manuscript is to obtain environmentally friendly composite materials basis on a polyurethane polymer from castor oil and clam shell powder as filler. The manuscript is addressed to readers interested in obtaining polymer materials from natural renewable resources and their characterization. The paper is written in four main parts including: introduction, experimental part, obtained results and final conclusions. At the beginning of the manuscript there is a short summary in which the authors make a brief presentation of the main aspects of the research initiated by the authors.
The summary is clearly written and provides a picture of the essence of the paper. However, I would recommend the authors to stop using abbreviations in this section because they make it difficult to follow the aspects presented in the work. Abbreviations can be used as needed in the other sections of the manuscript with the meaning defined immediately after use.
Answer : Thank you for your suggestion, we have updated this section and verified the abbreviations as well.
In the introduction section, the authors critically analyze the literature of recent years dedicated to the use of castor oil together with different diisocyanates and diols to obtain polyurethanes with improved flexibility. At the same time, the negative effects of the presence of castor oil on the other mechanical properties are highlighted, which limits the use of these polymers in structural applications; A simple solution taken from specialized literature to improve the mechanical characteristics of polymers is the creation of composite materials reinforced with inorganic particles such as calcium carbonate, barium carbonate, silica, etc. Carefully following the published data, the authors found that the presence of CaCO3 particles improves the mechano-dynamic and thermal properties of polymer composites. Starting from the presented aspects, the authors proposed in this paper to obtain composite materials from thermocrosslinkable polyurethane matrix based on castor oil and calcium carbonate particles of organic origin, such as the powder of clam shells. Although the idea of ​​using the powder from clam shells to obtain composite materials together with synthetic polymers is not really new, this approach is still interesting.
The second section of the manuscript is dedicated to experimental aspects. Both the materials used and the laboratory and the techniques for the preparation of the polyurethane prepolymer with castor oil content, the shell powder, as well as the preparation several composite made with the two types of materials are clearly described. The equipment used for characterization is quite well presented.
Data regarding the structural characterization of the obtained composites as well as the properties such as crystallinity, the behavior of the composites under mechanical stress, the morphological aspects are well described in the "Results" section. The thermograms in Figure 5 should also be completed with the derived curves. The stages of thermal degradation will be better highlighted. Other data such as the starting temperature of the thermal degradation (T5%), the temperature at which the thermal degradation occurs at the maximum speed (Tm) and the temperature that characterizes the end of the thermal degradation (Tf) also be filled in the manuscript and commented according with the filler content. The conclusions are well supported by experimental data. The selected bibliography supports the authors' approach regarding the chosen research direction. The publications are up-to-date and an exaggerated number of self-citations were not found.
Answer : Thank you for your valuable suggestion. In this instance, we did not list the changes but highlighted them in red within the revised document. It should be noted that we have supplemented the infrared spectra of shell powder and CaCO3 and the XRD pattern of shell powder, while the infrared of TCOPU is provided as supporting material.

Round 2
Reviewer 1 Report
Comments and Suggestions for Authors
All concerns were appropriately addressed